# Efficacy and safety of extracorporeal shockwave therapy for the treatment of chronic non-bacterial prostatitis: A systematic review and meta-analysis

**Ponco Birowo** *, **Ervandy Rangganata**, **Nur Rasyid**, **Widi Atmoko**

Department of Urology, Faculty of Medicine Universitas Indonesia, Cipto Mangunkusumo General Hospital, Jakarta, Indonesia

☯ These authors contributed equally to this work.

* ponco.birowo@gmail.com

**Data Availability Statement:** All relevant data are within the manuscript and its Supporting Information files.

## Abstract

### Purpose

Chronic pelvic pain syndrome (CPPS) is one of the most common outpatient urological diagnoses, and its incidence is increasing. Extracorporeal shockwave therapy (ESWT) has been suggested for relieving local perineal symptoms associated with chronic prostatitis/CPPS. Despite several treatment methods, no causal or standardized treatment is available for CPPS. This study aimed to investigate the efficacy and safety profile of ESWT for the treatment of chronic non-bacterial prostatitis.

### Materials and methods

Studies were collected using four search engines (Pubmed, Cochrane, ScienceDirect, and EBSCOHost), on May 16, 2020; and assessed based on predetermined inclusion and exclusion criteria. Two reviewers performed study selection. Studies were then analyzed using Review Manager 5.3 for the meta-analysis.

### Results

Seventy-four publications were initially retrieved, and three studies were considered for both qualitative and quantitative analyses. From these studies, we found that the use of ESWT was significantly associated with decreased pain domain (mean difference: -3.93; 95% confidence interval [CI] -5.13, -2.73; p<0.001), improved urinary score (mean difference: -1.79; 95% CI -2.38, -1.21; p<0.001), improved quality of life (mean difference: -1.71; 95% CI -2.12, -1.31; p<0.001), and improved National Institutes of Health chronic prostatitis symptom index (NIH-CPSI) score (mean difference: -5.45; 95% CI -5.74, -5.16; p<0.001) after 12 weeks of treatment.

### Conclusion

ESWT is efficacious and safe in reducing pain and improving urinary condition, NIH-CPSI score, and quality of life in patients with chronic non-bacterial prostatitis.

**Funding:** Ponco Birowo NKB-1520/UN2.RST/
HKP.05.00/2020 International Publication
Research Grant Universitas Indonesia www.ui.ac.
id.

**Competing interests:** No authors have competing
interests.

## Introduction

Since the acceptance of the consensus of Chronic Prostatitis/Chronic Pelvic Pain Syndrome (CP/CPPS) in 1995 [1], reports of this urological condition have increased to date [1–3]. The prevalence of CPPS from a recent study in Chinese males over 40 years of age is around 25% (1,091 out of 4315) [3]. Symptoms of CPPS include urinary and erectile dysfunctions and pain in the prostate, perineal, inguinal, scrotal, and suprapubic regions, lasting for at least 3 of the previous six months [1]. CPPS also affects patients' quality of life (QOL) due to urinary and erectile dysfunctions [4]. Worse, QOL is associated with tremendous pain and urinary symptoms. The disease's typical restrictions are pain sensations located in the prostate, testes, groin, back, pelvic floor, and suprapubic region [4].

To date, the pathophysiology of CPPS has not been completely understood [5]. Psychiatric and somatic factors have possible roles in its pathophysiology [5, 6]. Nevertheless, no infection or bacterial pathogen has been detected. Associations of previous infections, pelvic floor hypertension, local chemical alterations, and perfusion disturbances in the pathophysiology of CPPS remain under discussion [5]. The role of prostate in CPPS is argued because women can also experience CPPS-like symptoms. In addition, the disease has dysfunctional effects, which are associated with myofascial pain syndrome along with a neurological component [7].

Several medical therapies are available to treat this disease. Analgesics, anti-inflammatory agents, antibiotics, α-receptor blockers, and 5-α reductase inhibitors have been used as single or combination therapies, with variable success rates [8]. Some alternative therapies include physiotherapy, trigger-point massage, electromagnetic treatment, acupuncture, rectal massage, hyperthermia, thermotherapy, balloon dilatation, laser coagulation, invasive neuromodulation, and intraprostatic injection of botulinum toxin A [8]. However, these therapies have not been uniformly successful in treating chronic prostatitis (CP). At present, no causal or standardized treatment is available [8, 9]. As stated in the guideline released by the European Association of Urology, patients with CPPS should be managed in a multispecialty and multidisciplinary environment, considering all their symptoms [9].

Orthopedic studies revealed that low-energy extracorporeal shockwave therapy (ESWT) successfully treats orthopedic pain syndromes, fractures, and wound healing disorders [10]. ESWT has been suggested for relieving local perineal symptoms associated with CP/CPPS [11]. Recent studies have investigated the efficacy of perineal ESWT in patients with CPPS. ESWT may reduce pain by several mechanisms [11]. The mechanisms through which ESWT alters pain have included interrupting the nerve impulse flow by hyperstimulating nociceptors, healing tissue by revascularization processes, and reducing muscle tone and spasticity [11–13]. The aforementioned mechanisms were hypothesized by previous studies on the use of ESWT in orthopedic pain syndromes [10]. The same mechanisms have been proposed as pain alteration mechanisms in which ESWT might benefit patients with CPPS.

The study aimed to investigate the efficacy and safety profile of ESWT for the treatment of chronic non-bacterial prostatitis. By far, this is the first systematic review and meta-analysis conducted to pool any finding on the efficacy of ESWT in treating chronic non-bacterial prostatitis.

## Material and methods

### Database searching and literature screening

Studies were selected using PubMed, ScienceDirect, Cochrane, and EBSCOhost on May 16, 2020. The exact keywords used were as follows: (extracorporeal shock wave therapy OR ESWT) AND (placebo OR sham) AND (prostatitis OR chronic prostatitis OR chronic

**Table 1. Database, search terms, and number of articles retrieved.**

| Database | Search strategy | Hits |
|---|---|---|
| PubMed | ((extracorporeal shockwave therapy [All Fields] OR low intensity extracorporeal shockwave therapy [All Fields] OR eswt [All Fields]) AND (prostatitis OR chronic prostatitis OR chronic non-bacterial prostatitis OR chronic abacterial prostatitis OR chronic pelvic pain syndrome OR CPPS)) | 14 |
| Cochrane | ((extracorporeal shockwave therapy OR low intensity extracorporeal shockwave therapy OR eswt) AND (prostatitis OR chronic prostatitis OR chronic non-bacterial prostatitis OR chronic abacterial prostatitis OR chronic pelvic pain syndrome OR CPPS)) | 1 |
| ScienceDirect | ((extracorporeal shockwave therapy OR low intensity extracorporeal shockwave therapy OR eswt) AND (prostatitis OR chronic prostatitis OR chronic non-bacterial prostatitis OR chronic abacterial prostatitis OR chronic pelvic pain syndrome OR CPPS)) | 45 |
| EBSCOhost | ((extracorporeal shockwave therapy OR low intensity extracorporeal shockwave therapy OR eswt) AND (prostatitis OR chronic prostatitis OR chronic non-bacterial prostatitis OR chronic abacterial prostatitis OR chronic pelvic pain syndrome OR CPPS)) | 14 |

nonbacterial prostatitis OR chronic abacterial prostatitis). The protocol was also registered under PROSPERO database (ID number: 187793).

All keywords used were searched for their respective synonyms using the MeSH thesaurus. This data searching process was not limited by the date of publication, and only full-text articles written in English were used (Table 1). Article selection was performed according to the search strategy recommended by the Preferred Reporting Items for Systematic Reviews and Meta-analysis. Only studies comparing ESWT and placebo for chronic non-bacterial prostatitis were assessed for further analysis. Non-human studies and non-placebo-controlled studies were excluded from the review. Data from all selected articles were extracted independently by four reviewers. Any disagreements were resolved by consensus.

## Study selection

All studies were manually screened for duplication. Duplicate-free articles underwent title and abstract examination based on predetermined inclusion and exclusion criteria. Study selection was performed by four reviewers (ER, PB, NR, and WD). In the event of disagreement, the consensus was achieved through discussion or third-party adjudication. Studies that fulfilled the inclusion and exclusion criteria underwent full-text review.

## Eligibility criteria

This is a systematic review and meta-analysis study. Table 2 shows information on patients, interventions, comparison, and outcomes. The data searching process was not limited by the date of publication, and only full-text articles in English were included.

## Types of studies

This review included all studies investigating the efficacy and safety profile of ESWT for the treatment of chronic non-bacterial prostatitis. Full-text studies with a clinical trial design were included. No date and language restrictions were applied.

**Table 2. Patient, intervention, comparison, and outcome.**

| Patients | Patients with chronic nonbacterial prostatitis based on NIH classification |
|---|---|
| Interventions | Extracorporeal shockwave therapy (ESWT) |
| Comparisons | Placebo/Sham |
| Outcome | Degree of pain using visual analog scale (VAS), urinary score, quality of life, NIH-developed chronic prostatitis symptom index (NIH-CPSI) |

## Data extraction

For every selected full-text article, we extracted the following data if available: patient demographics, types of procedures (ESWT), number of sessions, the degree of pain, CPPS-related complaints, micturition conditions, and erectile function. Data collected were relevant information about interventions, characteristics, and outcomes that suited the inclusion criteria formed by the reviewers. The outcome measures were the degree of pain, which was evaluated using a visual analog scale (0–10), CPPS-related complaints, which were investigated using the National Institutes of Health chronic prostatitis symptom index (NIH-CPSI), urinary score, and QOL.

## Statistical analysis

Two independent reviewers conducted data analyses. Studies were assessed based on the Oxford Center of Evidence-Based Medicine Worksheet for therapy and analyzed using Review Manager 5.3 for the meta-analysis. Weighted mean differences and odds ratio were used to analyze the variables of each study. The confidence interval (CI) was 95%, and p-values less than 0.05 were considered significant.

Cochrane Q test was performed to assess the heterogeneity of the studies. Heterogeneity was evaluated using $I^2$ statistics; a value less than 50% indicated homogeneous studies, and a fixed effects model was used. If the value was more than 50%, studies were considered heterogeneous, and a random effects model was used.

# Results

A total of 74 publications were initially retrieved (Fig 1). Of these, 49 studies were excluded due to duplication, resulting in 15 studies for screening titles and abstracts. Ten studies that met the exclusion criteria were further excluded, resulting in 5 studies for full-text assessment. Furthermore, two studies were excluded because they were follow-up studies, and ESWT was performed in adjunct to medical therapies. Three articles were included for both qualitative and quantitative analyses.

## Study characteristics and quality

Characteristics of the included publications were listed in Table 3, and the summary of the results is presented in Table 4. A total of three studies were included in this analysis. The current meta-analysis only selected randomized clinical studies because they are considered to have the best study design for an interventional study. Quality of the studies were assessed by Jadad scale for randomized controlled trial and was presented in Table 3, while Cochrane Risk of Bias Tool was used to assess the risk of bias. According to Hartling et al. (2011), a study scored less than three by the Jadad scale was considered a low-quality study; thus, out of three included studies, two of those were considered a good quality study [14]. Overall, the risk of bias of included studies was considered low, as presented in Fig 2. Some parameters were still questioning, as several studies did not mention allocation concealment and blinding.

## Synthesis of results

The majority of the studies reviewed in this analysis had a low risk of bias (Fig 2). The study conducted by Moayednia et al. recruited 40 patients who had been randomly assigned into two groups: the treatment and sham groups. Patient selection was performed by choosing from a pool of patients diagnosed with CP type IIIB according to NIH classification. Blinding

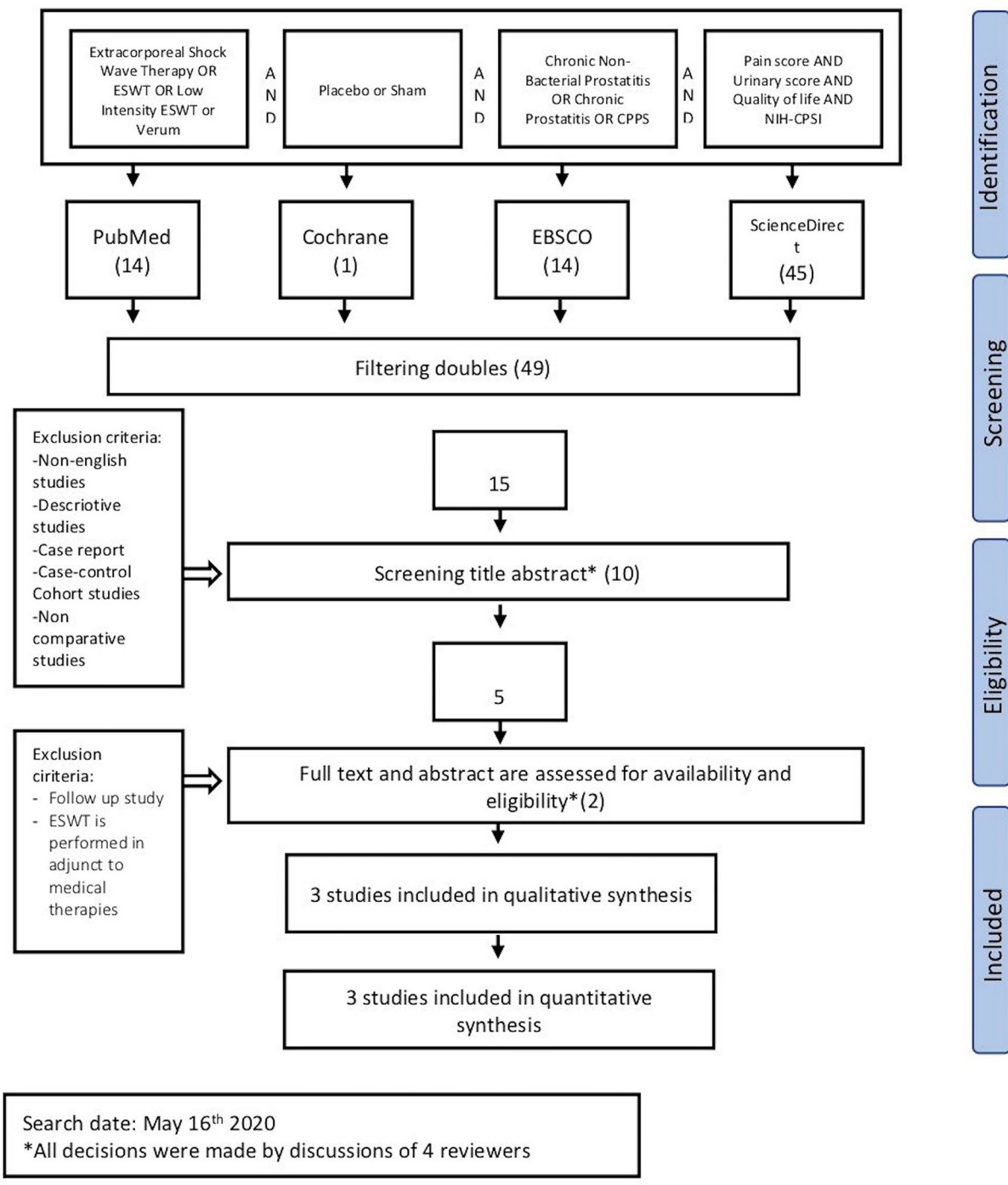

**Fig 1. Study flow diagram.**

was performed by adhering to the same protocol used in the treatment group with the probe turned off. The good and bad outcomes were reported impartially between the treatment and sham groups. Vahdatpour et al. (2013) and Zimmerman et al. (2009) used the same protocol that Moayednia et al. (2014) had.

**Table 3. Characteristics of the included studies.**

| Author(s) | Year | Study design | Number of patients | | Pain score | | Urinary score | | Quality of life | | NIH-CPSI | | IIEF | | IPSS | | Jadad scale |
|---|---|---|---|---|---|---|---|---|---|---|---|---|---|---|---|---|---|
| | | | ESWT | Control | ESWT | Control | ESWT | Control | ESWT | Control | ESWT | Control | ESWT | Control | ESWT | Control | |
| Moayednia et al. [15] | 2014 | Randomized, controlled trial | 19 | 18 | Baseline: 13.05 ± 2.60 Wk12: 9.15 ± 0.92 | Baseline: 13.77 ± 1.90 Wk12: 13.89 ± 1.47 | Baseline: 4.71 ± 2.69 Wk12: 3.68 ± 1.29 | Baseline: 5.19 ± 1.77 Wk12: 5.47 ± 1.18 | Baseline: 8.18 ± 1.71 Wk12:6.06 ± 0.72 | Baseline: 8.22 ± 2.20 Wk12: 7.79 ± 1.15 | Baseline: 26.03 ± 3.72 Wk12: 19.74 ± 1.65 | Baseline: 27.18 ± 2.51 Wk12: 26.81 ± 2.91 | N/A | N/A | N/A | N/A | 2 |
| Vahdatpour et al. [16] | 2013 | Randomized, controlled Trial | 20 | 20 | Baseline: 13.8 ± 2.6 Wk12: 9.5 ± 0.9 | Baseline: 13.6 ± 2 Wk12: 13.7 ± 1.6 | Baseline: 4.6 ± 2.8 Wk12: 3.7 ± 1.5 | Baseline: 5.2 ± 2 Wk12: 5.4 ± 1.3 | Baseline: 8.1 ± 1.7 Wk12: 6.1 ± 0.8 | Baseline: 8.3 ± 1.9 Wk12: 7.8 ± 0.9 | Baseline: 26.5 ± 3.4 Wk12: 19.4 ± 1.4 | Baseline: 27.1 ± 3.1 Wk12: 26.9 ± 3 | N/A | N/A | N/A | N/A | 3 |
| Zimmermann et al. [18] | 2009 | Randomized, double-blind, placebo-controlled study | 30 | 30 | Baseline: 5.33 ± 0.26 Wk12: 3.13 ± 0.28 | Baseline: 5.73 ± 0.20 Wk12: 6.13 ± 0.26 | N/A | N/A | N/A | N/A | Baseline: 23.20 ± 0.66 Wk12: 19.70 ± 0.67 | Baseline: 25.07 ± 0.48 Wk12: 25.00 ± 0.50 | Baseline: 18.27 ± 0.60 Wk12: 20.17 ± 0.32 | Baseline: 17.13 ± 0.68 Wk12: 16.83 ± 0.59 | Baseline: 15.83 ± 0.39 Wk12: 12.53 ± 0.31 | Baseline: 16.10 ± 0.38 Wk12: 17.03 ± 0.55 | 5 |

ESWT, extra corporeal shock wave therapy; NIH-CPSI, national institute of health chronic prostatitis symptom index; IIEF, international index of erectile function; IPSS, international prostate symptom score

**Table 4. Summary of the included studies (dose-response gradient).**

| Authors | Year | Study design | Protocol | Sham group | Evaluation | Results |
|---|---|---|---|---|---|---|
| Moayednia et al. [15] | 2014 | Randomized, controlled Trial | 4 weekly sessions 3000 SW, *Efd* 0.25 mJ/mm² (increased 0.05 mJ/mm² each week), 3 Hz Standard electromagnetic DUOLITH SD1, Storz Medical, Tägerwilen, Switzerland | The same protocol was used. However, the probe was turned off. | First week to week 12, until 24 weeks post-treatment | All four domains (pain domain, urinary score, QoL, NIH-CPSI) were statistically different at weeks and week 12. At week 24, no statistical difference was found in pain score, urinary score, QoL, and NIH-CPSI score between the two groups. |
| Vahdatpour et al. [16] | 2013 | Randomized, controlled trial | 4 weekly sessions 3000 SW, *Efd* 0.25 mJ/mm² (increased 0.05 mJ/mm² each week), 3 Hz Standard electromagnetic DUOLITH SD1, Storz Medical, Tägerwilen, Switzerland. | The same protocol was used. However, the probe was turned off. | 1, 2, 3, and 12 weeks following the first ESWT | • Pain domain scores were statistically significant after the second treatment session. The scores were reduced in the treatment and sham groups.<br>• Urinary score was statistically different at weeks 3 and 12 between the treatment and sham groups.<br>• QOL decreased more significantly at all four follow-up time points in the treatment group.<br>• NIH-CPSI scores decreased more significantly at all four follow-up time points in the treatment group.<br>• The outcomes of this study in the treatment group were improved during the 3-week treatment. A slight deterioration was observed at week 12 of the follow-up. |

(*Continued*)

Table 4. (Continued)

| Authors | Year | Study design | Protocol | Sham group | Evaluation | Results |
|---|---|---|---|---|---|---|
| Zimmermann et al. [18] | 2009 | Randomized, double-blind, placebo-controlled study | 4 weekly sessions 3000 SW, *Efd* 0.25 mJ/mm2, 3 Hz. The focus zone penetration depth was in the range of 35–65 mm. Standard electromagnetic DUOLITH SD1, Storz Medical, Tägerwilen, Switzerland. | The placebo group received a placebo stand-off, which contained a shockwave-absorbing material, a layer of air, and air-filled microspheres. | 1, 4, and 12 weeks following ESWT | All 30 patients in the treatment group improved significantly in pain, quality of life, and voiding conditions compared with the placebo group. The placebo group experienced some deterioration during the follow-up period. |

## Pain domain

The baseline pain score was slightly lesser in the treatment group (mean difference: -0.40; 95% CI -0.51, -0.28; p<0.001; Fig 3, above). After 12 weeks of treatment, pooled analysis showed that ESWT significantly reduced the pain score (mean difference: -3.93; 95% CI -5.13, -2.73; p<0.001; Fig 3, below).

## Urinary score

The baseline characteristics of the urinary score showed no statistically significant difference (mean difference: -0.54; 95% CI -1.59, 0.51; p = 0.31; Fig 4, above). Improvement was observed in the urinary score after 12 weeks of treatment (mean difference: -1.79; 95% CI -2.38, -1.21; p<0.001; Fig 4, below).

## Quality of life

The QOL parameter was assessed in the studies conducted by Moayednia et al. (2014) and Vah-datpour et al. (2013). The studies that indicated an $I^2$ statistic of 0% were not heterogenous. The baseline characteristics of the parameter was unremarkable (mean difference -0.13; 95% CI -0.97, 0.71; p = 0.76; Fig 5, above). The studies revealed that ESWT improved the QOL after 12 weeks of treatment (mean difference: -1.71; 95% CI -2.12, -1.31; p<0.001; Fig 5, below).

## NIH-CPSI score

The NIH-CPSI score was not comparable between the groups at the beginning of the study. Fig 6 (above) shows that the baseline value of NIH-CPSI score was different between the ESWT and control groups (mean difference: -1.83; 95% CI -2.12, -1.54; p<0.001). Twelve weeks after treatment, the difference became more distinct. Compared with the control group, ESWT improved the NIH-CPSI score in patients with chronic non-bacterial prostatitis by 5.45 points (95% CI -5.74, -5.16; p<0.001; Fig 6, below). The studies of NIH-CPSI Score in twelve weeks were heterogenous ($I^2$ = 85%).

## Discussion

The present meta-analysis investigated the efficacy and safety profile of ESWT in the manage-ment of chronic non-bacterial prostatitis based on NIH classification. A randomized clinical

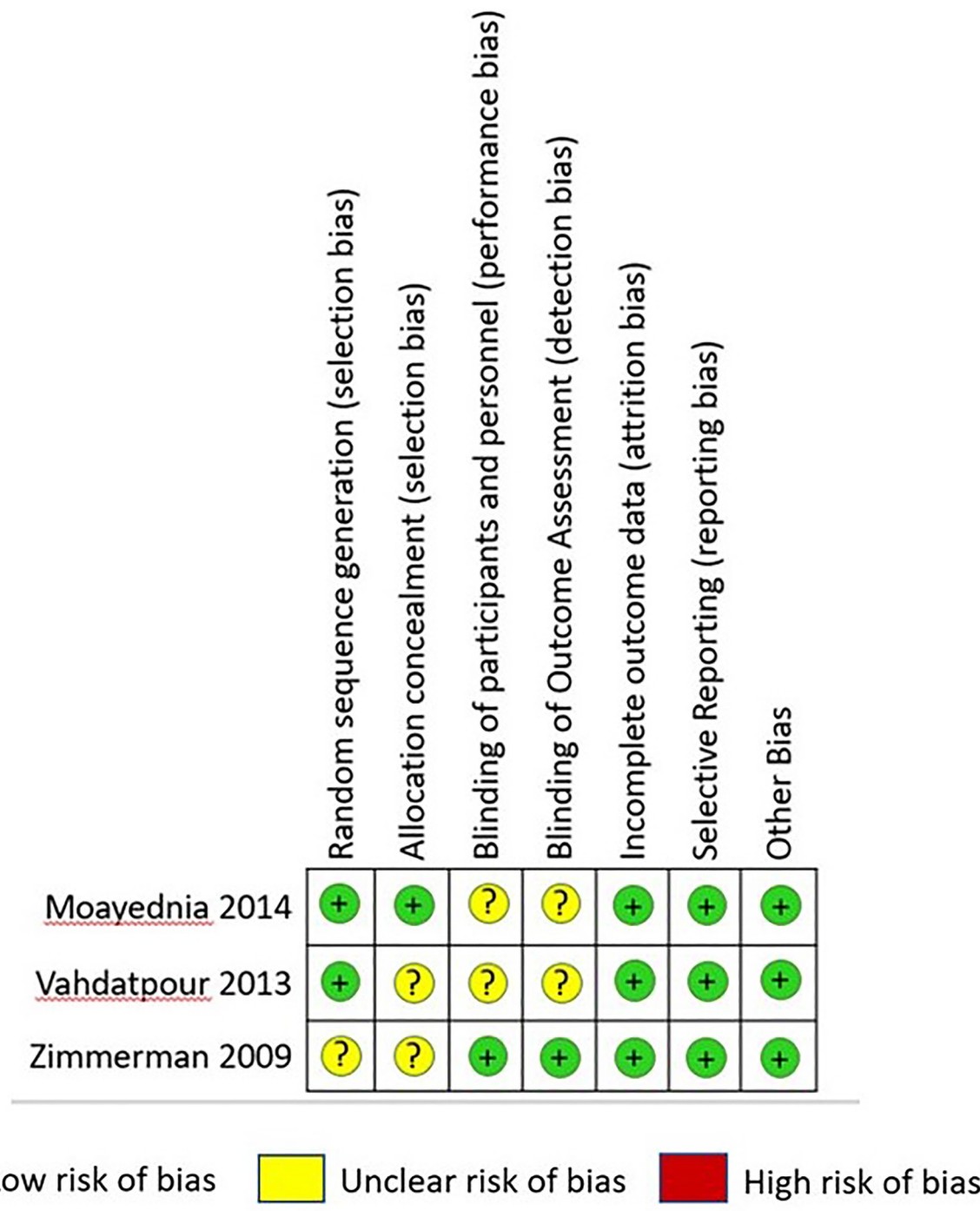

**Fig 2. Risk of bias summary.**

trial is the best study design to evaluate such type of study. This meta-analysis included three randomized clinical trial studies, all of which were conducted with a treatment arm and a sham arm. We found that ESWT seemed to be safe and effective for the treatment of chronic non-bacterial prostatitis. The pooled mean differences for pain domain, urinary score, QOL, and NIH-CPSI score were all statistically significant. This finding indicates that the use of

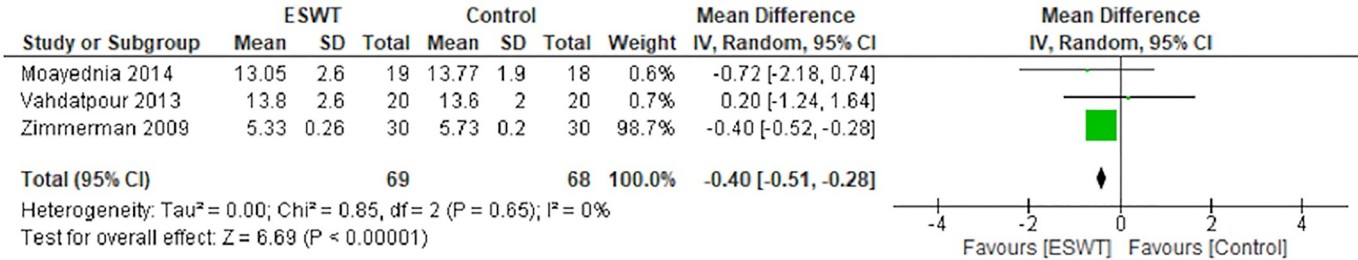

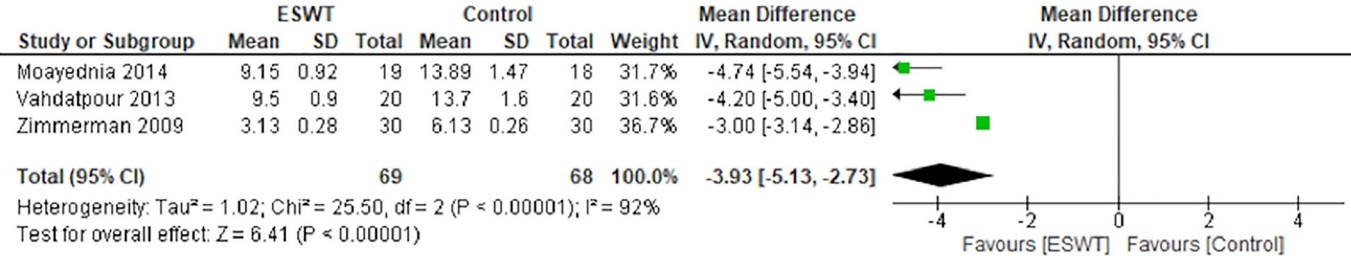

**Fig 3. Forest plot of baseline pain domain (above) and after 12 weeks (below).**

ESWT reduces pain and improves urinary score, QOL, and CP symptoms. The first study conducted by Zimmermann et al. showed that ESWT resulted in statistically significant improvements in pain and QOL, whereas voiding conditions, which were measured using the international prostate symptom score (IPSS), also improved but without statistical significance [15]. In the included cohort, an increase in serum prostate-specific antigen was found in 17% of patients two days after treatment [16]. In other studies, an increase of less than 10% or even a decrease was observed [17]. That study revealed that ESWT does not appear to be traumatic for the prostate gland. No pain or discomfort was observed during or after the procedure [17].

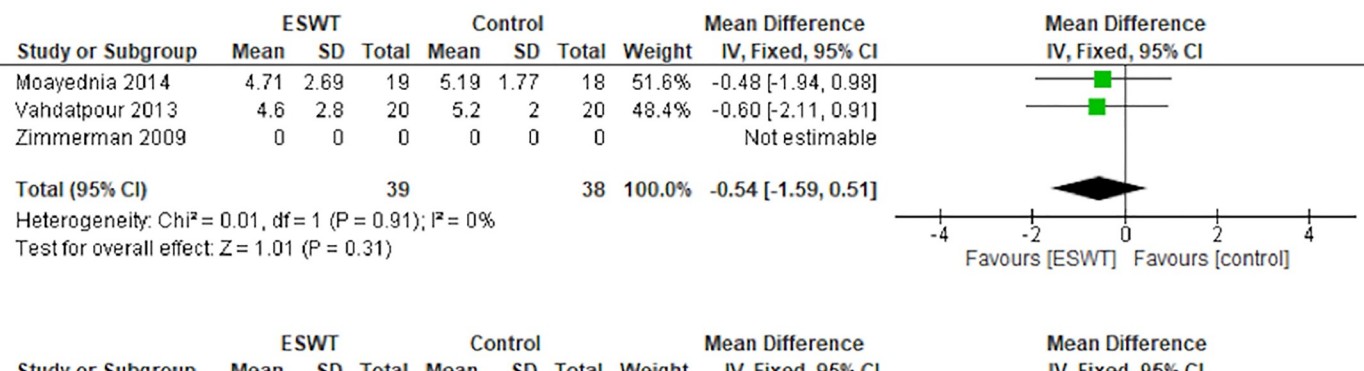

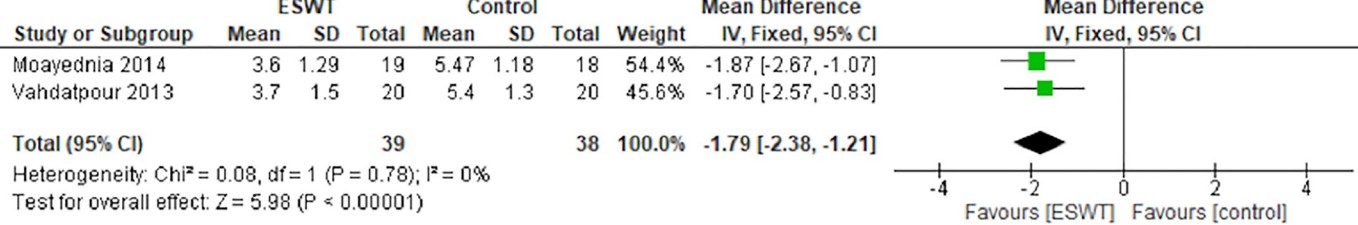

**Fig 4. Forest plot of baseline urinary score (above) and after 12 weeks (below).**

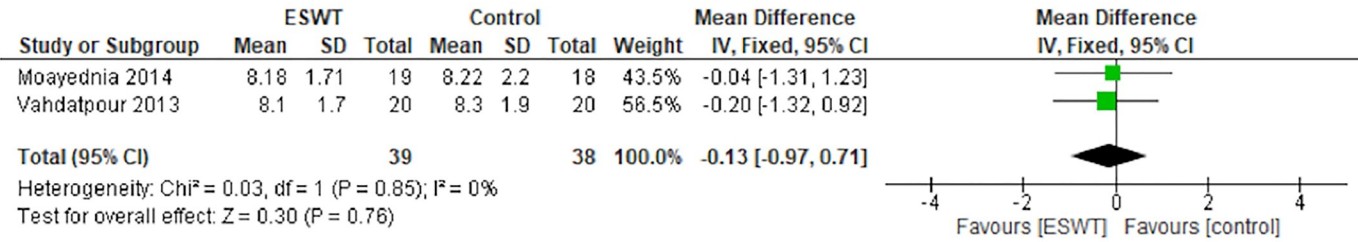

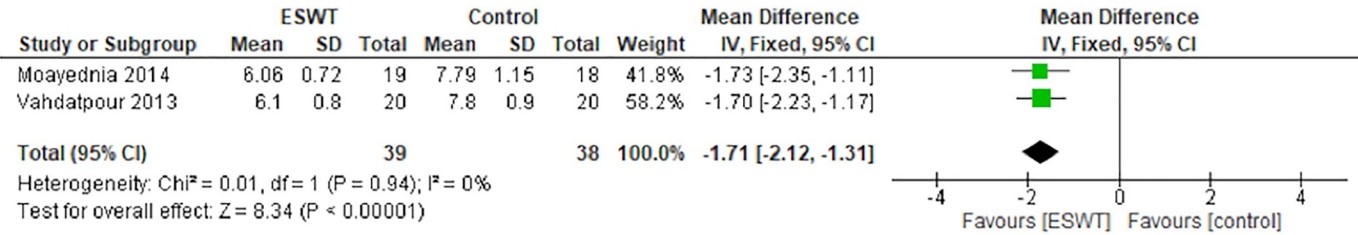

**Fig 5. Forest plot of baseline quality of life (above) and after 12 weeks (below).**

Zimmermann et al. [18] conducted a similar trial that included 60 patients. They used the NIH-CPSI, IPSS, international index of erectile function, and visual analog scale to investigate their parameters. Zimmermann et al. had a greater proportion of patients who had undergone ESWT treatment. Those patients showed reduced pain and improved QOL, whereas those in the control group presented no improvement [18].

All studies reported significantly lower 12-week pain scores in the treatment group. On long-term observation, Moayednia et al. [15] found that the pain score difference gradually diminished and became insignificant by week 24. Currently, no other studies have reported pain scores in the long-term observation period. Therefore, clinical trials with more extended periods are deemed necessary to evaluate the long-term efficacy of ESWT in patients with chronic non-bacterial prostatitis.

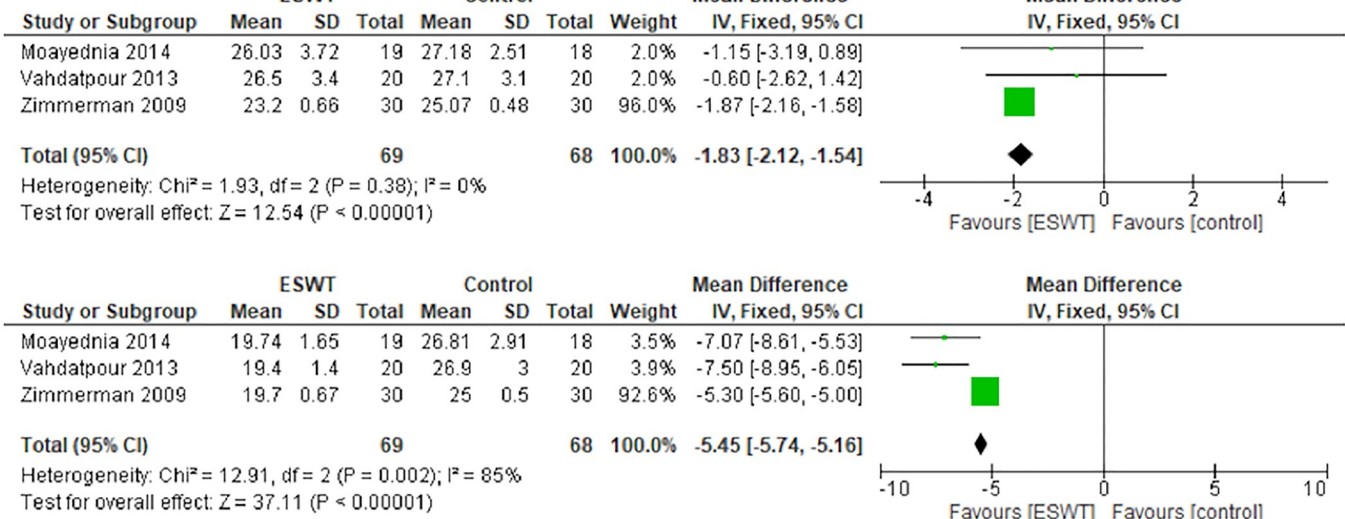

**Fig 6. Forest plot of baseline NIH-CPSI score (above) and after 12 weeks (below).**

The pathogenesis of chronic non-bacterial prostatitis is not completely understood [19]. The theories describing the mechanisms of the disease have included infection, which leads to pain via nociceptive nerve endings and receptors, pelvic floor hyperactivity, local chemical alterations, neurologic components, and perfusion disturbances [6, 19]. The role of the prostate remains questionable. Extracorporeal shockwaves affect the tissue by transforming the mechanical signals into biochemical or molecular biologic signals [20, 21]. The modulation of pain signal transmission may occur by producing extracellular cavitation, by activating the small-diameter fibers and the serotonergic system, and by applying the gate-control theory [20, 21]. Extracellular cavitation may be produced by the waves as they pass through human tissues. As a result, cavitation damages the local nerve endings. Although no consensus exists about the mechanism of ESWT on CPPS, some considerations that can be used to alter the pain in CP management include reducing passive muscle tone, hyperstimulating nociceptors, interrupting the flow of nerve impulses, and influencing the neuroplasticity of the pain memory [18].

Several studies have shown that ESWT has low adverse effects. A study conducted by Zimmerman et al. revealed the absence of prostate-specific antigen rise after ESWT, which confirmed that ESWT has low adverse effects [18].

A conclusion of the dose-response gradient analysis could not be obtained from the included studies because all studies conducted four weekly ESWT sessions (3000 SW, Efd 0.25 mJ/mm$^2$ [increased 0.05 mJ/mm$^2$ each week], 3 Hz). All studies utilized the standard electromagnetic DUOLITH SD1 (Storz Medical, Tägerwilen, Switzerland). Vahdatpour et al. implemented a different method by adding 0.5 mJ/mm$^2$ in each week of the treatment.

The ESWT effect was shown to be dose dependent. In a study conducted by Vahdatpour et al., the numbers of shockwaves and the energy level were empirical. In their study, the selection of the number of treatments, the treatment intervals, and the number of pulses per session was made according to clinical studies on previous application. A modification was made by adding 0.5 mJ/mm$^2$ in each week. Patients showed improvement in their symptoms in week 3 compared with week 2. The improvement did not continue until week 12 [16]. As a result, a definitive conclusion could not be drawn for the long-term effect of this study protocol.

Multimodal therapy for CPPS has been proposed for a long time and became famous when a study reported the CPPS monotherapy strategy's failure [22]. Multimodal therapy was proposed as CPPS presents as a disease entity with a complex etiology and pathogenesis, so regimens combining alpha-blockers and antibiotics are recommended for patients [23]. Given the rationalization for multimodal therapy and considering that ESWT has good efficacy in improving CPPS patients' symptoms, clinical studies involving ESWT in multimodal therapy, or those comparing ESWT in multimodal therapy versus pre-existing multimodal therapy without ESWT, are deemed essential.

The present study has several strengths and limitations. All included studies had low risk of bias, and, overall, a low heterogeneity existed among the studies, except for the pain domain. However, to date, no consensus has been reached on the standardized dose of ESWT. Two of the included studies modified the ESWT dose by adding 0.5 mJ/mm$^2$ in each week of the treatment. In addition, the outcomes of the included studies were different in some domains. Heterogeneity assessment was likely to be biased in this meta-analysis. According to von Hippel (2015), using either I2 or Cochrane Q to assess the heterogeneity in the small meta-analysis could possibly be biased. The desirable choice was to supplement or replace those parameters with a confidence interval [24]. However, this could not completely address the heterogeneity prediction problem in this small meta-analysis. Further studies about the pathophysiology of CP are needed to better explain the pathway in which ESWT benefits patients with CP/CPPS.

## Conclusion

The present meta-analysis found that compared with the placebo/sham group, ESWT is efficacious and safe in reducing pain and improving the urinary condition, NIH-CPSI score, and QOL in patients with chronic non-bacterial prostatitis. The obtained results of this study can be potentially applied into the pre-existing practical evidence-based guideline to treat chronic non-bacterial prostatitis. Nonetheless, further investigation with long-term follow-up is essential to describe a standard protocol for ESWT.

## Supporting information

**S1 Checklist. PRISMA 2009 checklist.**
(PDF)

## Acknowledgments

The authors would like to thank Cipto Mangunkusumo General Hospital for the support in this study.

## Author Contributions

**Conceptualization:** Ponco Birowo, Nur Rasyid, Widi Atmoko.

**Data curation:** Ponco Birowo, Ervandy Rangganata, Widi Atmoko.

**Formal analysis:** Ponco Birowo, Ervandy Rangganata.

**Funding acquisition:** Ponco Birowo.

**Investigation:** Ponco Birowo, Ervandy Rangganata.

**Methodology:** Ponco Birowo, Ervandy Rangganata.

**Resources:** Widi Atmoko.

**Software:** Widi Atmoko.

**Supervision:** Ponco Birowo, Nur Rasyid, Widi Atmoko.

**Validation:** Ponco Birowo, Nur Rasyid, Widi Atmoko.

**Visualization:** Ponco Birowo, Nur Rasyid, Widi Atmoko.

**Writing – original draft:** Ponco Birowo, Ervandy Rangganata, Nur Rasyid, Widi Atmoko.

**Writing – review & editing:** Ponco Birowo, Ervandy Rangganata, Nur Rasyid, Widi Atmoko.

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
