## [Decision Letter · Decision Letter 0]

16 Oct 2020

PONE-D-20-28346

Efficacy and safety of extracorporeal shockwave therapy for the treatment of chronic non-bacterial prostatitis: a systematic review and meta-analysis

PLOS ONE

Dear Dr. Birowo,

Thank you for submitting your manuscript to PLOS ONE. After careful consideration, we feel that it has merit but does not fully meet PLOS ONE’s publication criteria as it currently stands. Therefore, we invite you to submit a revised version of the manuscript that addresses the points raised during the review process.

We look forward to receiving your revised manuscript.

Kind regards,

Marco Aurélio Gouveia Alves

Academic Editor

PLOS ONE

Journal Requirements:

3. Thank you for submitting the above manuscript to PLOS ONE. During our internal evaluation of the manuscript, we found significant text overlap between your submission and the following previously published works.

- https://www.hindawi.com/journals/isrn/2013/972601/

Please revise the manuscript to rephrase the duplicated text, cite your sources, and provide details as to how the current manuscript advances on previous work. Please note that further consideration is dependent on the submission of a manuscript that addresses these concerns about the overlap in text with published work.

Reviewers' comments:

Reviewer's Responses to Questions

**Comments to the Author**

1. Is the manuscript technically sound, and do the data support the conclusions?

Reviewer #1: Partly

Reviewer #2: Partly

2. Has the statistical analysis been performed appropriately and rigorously? 

Reviewer #1: Yes

Reviewer #2: Yes

3. Have the authors made all data underlying the findings in their manuscript fully available?

Reviewer #1: Yes

Reviewer #2: Yes

4. Is the manuscript presented in an intelligible fashion and written in standard English?

Reviewer #1: Yes

Reviewer #2: No

5. Review Comments to the Author

Reviewer #1: In this study, the authors aimed to assess the efficacy and safety profile of ESWT for the treatment of chronic non-bacterial prostatitis. After reading this manuscript, I've found a few aspects that are a matter of concern. Very few studies have been considered to support the final assumptions concerning the consensus on a dose of ESWT. The authors should also take some caution regarding Pain Domain conclusions.

Reviewer #2: This systematic review and meta-analysis aims to demonstrate the efficacy and safety of ESWT for the treatment of chronic non-bacterial prostatitis. In general, the English language should be improved and errors corrected. After their research, authors analyzed the results of 3 papers only. Discussion does not add relevant data to what is mentioned in results. Other comments:

Abstract - Please detail some search engines used in this MS and period of study

Keywords: Please add “Chronic pelvic pain syndrome”

Introduction: This section must be more robust, making an upto date of the literature. In addition, please detail and rephrase aims of study, since some papers are available on ESWT treatment.

A quick search showed the following papers:

Polackwich AS, Shoskes DA. Chronic prostatitis/chronic pelvic pain syndrome: a review of evaluation and therapy. Prostate Cancer Prostatic Dis. 2016 Jun;19(2):132-8. doi: 10.1038/pcan.2016.8.

Clemens JQ, et al., Research Network Study Group. Urologic chronic pelvic pain syndrome: insights from the MAPP Research Network. Nat Rev Urol. 2019 Mar;16(3):187-200. doi: 10.1038/s41585-018-0135-5.

DeWitt-Foy ME et al., Management of Chronic Prostatitis/Chronic Pelvic Pain Syndrome. Eur Urol Focus. 2019 Jan;5(1):2-4. doi: 10.1016/j.euf.2018.08.027.

Magistro G, et al., Contemporary Management of Chronic Prostatitis/Chronic Pelvic Pain Syndrome. Eur Urol. 2016 Feb;69(2):286-97. doi: 10.1016/j.eururo.2015.08.061.

Rayegani SM, et al., Extracorporeal Shockwave Therapy Combined with Drug Therapy in Chronic Pelvic Pain Syndrome : A Randomized Clinical Trial. Urol J. 2020 Mar 16;17(2):185-191. doi: 10.22037/uj.v0i0.4673.

Zhang ZX, et al. Efficacy of Radial Extracorporeal Shock Wave Therapy for Chronic Pelvic Pain Syndrome: A Nonrandomized Controlled Trial. Am J Mens Health. 2019 Jan-Feb;13(1):1557988318814663. doi: 10.1177/1557988318814663.

Database Searching and Literature Screening: The searching process of these data ….

Results: Is this analysis based on three studies only? Which parameters were used to qualify two of those studies as good quality?

Table 3 is expendable, since data are included in Table 4. Please correct errors in the footer of Table 4.

Discussion: Please start discussion section with the paragraph: “The present meta-analysis investigated the efficacy and safety profile of ESWT in the management of chronic non-bacterial prostatitis based on NIH classification. A randomized clinical trial is the best study design to evaluate such type of study. This meta-analysis included three randomized clinical trial studies, all of which were conducted with a treatment arm and a sham arm”.

Please include references on non existed consensus exists on the mechanism of ESWT on CPPS.

Please discuss other therapeutic startegies since a number of studies suggest a multimodal therapeutic approach.

6. PLOS authors have the option to publish the peer review history of their article (what does this mean?). If published, this will include your full peer review and any attached files.

Reviewer #1: No

Reviewer #2: **Yes: **Maria de Lourdes Pereira

---

## [Author Response · Author response to Decision Letter 0]

2 Nov 2020

Response to Reviewers

Submission ID: PONE-D-20-28346

Title : Efficacy and safety of extracorporeal shockwave therapy for the treatment of chronic non-bacterial prostatitis: a systematic review and meta-analysis

Dear Editor, Reviewer 1, and Ms./Mrs. Maria de Lourdes Pereira

Thank you for taking the time to read our article and provide feedback on improving it. We have made quite a lot of changes in our article, and these are some point we would like to enclose regarding the comment that editor and reviewers have given:

Editor:

“Please ensure that your manuscript meets PLOS ONE's style requirements, including those for file naming. The PLOS ONE style templates can be found at…”

According to the template given, we have made an update regarding our article style, including the file naming.

“Please include captions for your Supporting Information files at the end of your manuscript, and update any in-text citations to match accordingly. Please see our Supporting Information guidelines for more information: http://journals.plos.org/plosone/s/supporting-information”

Thank you for the suggestion given. We thought that the pictures and tables in our main article are sufficient to support our article; thus, we neither consider adding additional figures nor tables as Supporting Information.

“Thank you for submitting the above manuscript to PLOS ONE. During our internal evaluation of the manuscript, we found significant text overlap between your submission and the following previously published works.”

We have updated the sections that overlap with the previously published article. We apologize for this negligence; we will try to be more thorough before submitting the article.

Reviewer 1:

“Very few studies have been considered to support the final assumptions concerning the consensus on a dose of ESWT.”

There are only a few studies available that discuss ESWT dosage. Thus, we also include this within the limitations of our study. The dose-response gradient cannot be assessed given that all included studies had the same dose of ESWT.

“The authors should also take some caution regarding Pain Domain conclusions.”

The pain domain after 12 weeks was the only outcome that had significant heterogeneity (92%). Therefore, we tried to analyze the effect size using the random-effects model. In this analysis, we found significantly lower results in the intervention group. However, this high heterogeneity may lead to less precise results, especially with the small number of studies in our meta-analysis. Thus, we also include this within the limitations of our meta-analysis.

Reviewer 2:

“In general, the English language should be improved and errors corrected.”

We would like to thank you for the feedback given. We have made several changes in our article. We also have tried our best to improve grammatical stuff in our article. We are looking forward for any comment regarding these revisions.

“Abstract - Please detail some search engines used in this MS and period of study

Keywords: Please add “Chronic pelvic pain syndrome”

Corrected

“Introduction - This section must be more robust, making an upto date of the literature. In addition, please detail and rephrase aims of study, since some papers are available on ESWT treatment.”

Changed

“Results: Is this analysis based on three studies only? Which parameters were used to qualify two of those studies as good quality? Table 3 is expendable, since data are included in Table 4. Please correct errors in the footer of Table 4.”

This analysis has made based on three studies available only. We used Jadad scale and Cochrane Risk Bias tools to assess the quality of these studies. We have expended the Table 3, and add a row in Table 4 informing the quality of study (Jadad scale score).

“Discussion: Please start discussion section with the paragraph: “………

Please discuss other therapeutic startegies since a number of studies suggest a multimodal therapeutic approach.”

Corrected. We also have added some discussion regarding the multimodal therapeutic strategies.

---

## [Decision Letter · Decision Letter 1]

2 Dec 2020

PONE-D-20-28346R1

Efficacy and safety of extracorporeal shockwave therapy for the treatment of chronic non-bacterial prostatitis: a systematic review and meta-analysis

PLOS ONE

Dear Dr. Birowo,

Thank you for submitting your manuscript to PLOS ONE. After careful consideration, we feel that it has merit but does not fully meet PLOS ONE’s publication criteria as it currently stands. Therefore, we invite you to submit a revised version of the manuscript that addresses the points raised during the review process.

We look forward to receiving your revised manuscript.

Kind regards,

Marco Aurélio Gouveia Alves

Academic Editor

PLOS ONE

Reviewers' comments:

Reviewer's Responses to Questions

**Comments to the Author**

1. If the authors have adequately addressed your comments raised in a previous round of review and you feel that this manuscript is now acceptable for publication, you may indicate that here to bypass the “Comments to the Author” section, enter your conflict of interest statement in the “Confidential to Editor” section, and submit your "Accept" recommendation.

Reviewer #1: All comments have been addressed

Reviewer #2: All comments have been addressed

2. Is the manuscript technically sound, and do the data support the conclusions?

Reviewer #1: Yes

Reviewer #2: Yes

3. Has the statistical analysis been performed appropriately and rigorously? 

Reviewer #1: Yes

Reviewer #2: Yes

4. Have the authors made all data underlying the findings in their manuscript fully available?

Reviewer #1: Yes

Reviewer #2: Yes

5. Is the manuscript presented in an intelligible fashion and written in standard English?

Reviewer #1: Yes

Reviewer #2: Yes

6. Review Comments to the Author

Reviewer #1: The revised version has been improved. From my point of view the explanation regarding pain domain should be in the manuscript.

Reviewer #2: The authors made a set of changes previously suggested, and also included some references, which improved the MS.

7. PLOS authors have the option to publish the peer review history of their article (what does this mean?). If published, this will include your full peer review and any attached files.

Reviewer #1: No

Reviewer #2: **Yes: **Maria de Lourdes Pereira

---

## [Author Response · Author response to Decision Letter 1]

6 Dec 2020

Response to Reviewers

Submission ID: PONE-D-20-28346

Title : Efficacy and safety of extracorporeal shockwave therapy for the treatment of chronic non-bacterial prostatitis: a systematic review and meta-analysis

Dear Editor, Reviewer 1, and Ms./Mrs. Maria de Lourdes Pereira

Thank you for taking the time to read our article and provide feedback on improving it. We have made quite a lot of changes in our article, and these are some point we would like to enclose regarding the comment that editor and reviewers have given:

Reviewer 1:

“The revised version has been improved. From my point of view the explanation regarding pain domain should be in the manuscript.”

Thank you for the feedback has been given. We have added a brief additional explanation regarding the pain domain that we have found in this study, specifically in the discussion section.

---

## [Editor Report · Decision Letter 2]

8 Dec 2020

Efficacy and safety of extracorporeal shockwave therapy for the treatment of chronic non-bacterial prostatitis: a systematic review and meta-analysis

PONE-D-20-28346R2

Dear Dr. Birowo,

We’re pleased to inform you that your manuscript has been judged scientifically suitable for publication and will be formally accepted for publication once it meets all outstanding technical requirements.

Kind regards,

Marco Aurélio Gouveia Alves

Academic Editor

PLOS ONE
---

## [Editor Report · Acceptance letter]

14 Dec 2020

PONE-D-20-28346R2 

Efficacy and safety of extracorporeal shockwave therapy for the treatment of chronic non-bacterial prostatitis: a systematic review and meta-analysis 

Dear Dr. Birowo:

I'm pleased to inform you that your manuscript has been deemed suitable for publication in PLOS ONE. Congratulations! Your manuscript is now with our production department. 

Kind regards, 

on behalf of

Dr. Marco Aurélio Gouveia Alves 

Academic Editor

PLOS ONE